# The Impact of Combined Nutrition and Exercise Interventions in Patients with Chronic Kidney Disease

**DOI:** 10.3390/nu16030406

**Published:** 2024-01-30

**Authors:** Manon de Geus, Manouk Dam, Wesley J. Visser, Karin J. R. Ipema, Anneke M. E. de Mik-van Egmond, Michael Tieland, Peter J. M. Weijs, Hinke M. Kruizenga

**Affiliations:** 1Department of Internal Medicine, Division of Dietetics, Erasmus MC, University Medical Center, 3015 GD Rotterdam, The Netherlands; w.j.visser@erasmusmc.nl (W.J.V.); a.vanegmond@erasmusmc.nl (A.M.E.d.M.-v.E.); 2Center of Expertise Urban Vitality, Faculty of Sports and Nutrition, Amsterdam University of Applied Sciences, 1067 SM Amsterdam, The Netherlands; michael.tieland@deakin.edu.au (M.T.); p.j.m.weijs@hva.nl (P.J.M.W.); h.kruizenga@amsterdamumc.nl (H.M.K.); 3Amsterdam UMC, Location Vrije Universiteit Amsterdam, Nutrition and Dietetics, Amsterdam Cardiovascular Sciences, De Boelelaan 1117, 1081 HV Amsterdam, The Netherlands; 4Department of Dietetics, University Medical Center Groningen, University of Groningen, 9713 GZ Groningen, The Netherlands; k.ipema@umcg.nl; 5Amsterdam Public Health, Aging and Later Life, 1081 HV Amsterdam, The Netherlands; 6Amsterdam Movement Sciences, Ageing and Vitality, 1081 HZ Amsterdam, The Netherlands

**Keywords:** chronic kidney disease, nutrition, muscle strength, quality of life, malnutrition

## Abstract

Combined nutrition and exercise interventions potentially improve protein-energy wasting/malnutrition-related outcomes in patients with chronic kidney disease (CKD). The aim was to systematically review the effect of combined interventions on nutritional status, muscle strength, physical performance and QoL. MEDLINE, Cochrane, Embase, Web of Science and Google Scholar were searched for studies up to the date of July 2023. Methodological quality was appraised with the Cochrane risk-of-bias tool. Ten randomized controlled trials (nine publications) were included (334 patients). No differences were observed in body mass index, lean body mass or leg strength. An improvement was found in the six-minute walk test (6-MWT) (*n* = 3, MD 27.2, 95%CI [7 to 48], *p* = 0.008), but not in the timed up-and-go test. No effect was found on QoL. A positive impact on 6-MWT was observed, but no improvements were detected in nutritional status, muscle strength or QoL. Concerns about reliability and generalizability arise due to limited statistical power and study heterogeneity of the studies included.

## 1. Introduction

Protein-energy wasting (PEW) and malnutrition are common in patients with chronic kidney disease (CKD), characterized by metabolic and nutritional alterations, leading to loss of muscle mass, muscle strength, physical activity, physical performance and quality of life (QoL) [1,2,3,4]. While the terms PEW and malnutrition are sometimes used interchangeably, malnutrition specifically refers to the loss of body weight, muscle and body fat resulting from inadequate nutrient intake and inflammation [5]. PEW provides a broader understanding of CKD-related mechanisms, including the hypercatabolic pathway driven by uremic toxins and inflammation [6]. Given that the term malnutrition is preferred by the Global Leadership Initiative on Malnutrition (GLIM) and PEW in the context of CKD, we will use the term PEW/malnutrition [7].

Nutritional strategies are important in preventing and reversing protein and energy depletion [6]. Since most patients with CKD experience strict nutritional restrictions and disease-related anorexia, it is often challenging to achieve an adequate nutrient intake [8]. Multiple studies investigated the effect of nutrition interventions whereby adequate amounts of protein and energy were provided. Some evidence suggests that these interventions may improve muscle protein synthesis and augment skeletal muscle mass [9,10,11,12].

Physical inactivity is common among patients with CKD and is associated with loss of muscle mass and poor physical performance [13,14].

In order to counteract these negative outcomes, various exercise regimens for patients with CKD were proposed. Indeed, studies have shown that specific types of exercise, such as aerobic or resistance training, improve muscle mass and physical performance in CKD patients [15,16]. Resistance-type exercise training increases leg muscle and hand grip strength (HGS) in patients with CKD, while regular exercise training is associated with improved health-related QoL in this population [16,17,18].

Combining nutrition and exercise interventions may be the most optimal strategy to increase the anabolic effect, as increased protein availability after exercise facilitates muscle amino acid uptake and improves muscle protein accretion [19,20,21,22,23]. In patients with CKD, combining protein supplementation with exercise enhances muscle amino acid uptake [24] and improves net muscle protein balance [25]. While these interventions appear promising for improving PEW/malnutrition-related outcomes in patients with CKD, the precise impact remains poorly understood. To address this knowledge gap, we aim to systematically review the evidence regarding the effect of combined interventions in patients with CKD on nutritional status, muscle strength, physical performance and QoL.

## 2. Materials and Methods

### 2.1. Study Protocol

This systematic review and meta-analysis were executed in accordance with Preferred Reporting Items for Systematic Reviews and Meta-Analyses (PRISMA) guidelines [26], and a pre-documented protocol was published (PROSPERO CRD42022358440).

### 2.2. Search Strategy

Database searches were performed in MEDLINE, the Cochrane Central Register of Controlled Trials, Embase, Web of Science and Google Scholar up to 31 July 2023. A specialized librarian was consulted for the completion of the finalized search string in which both MeSH and free text terms were used.

### 2.3. In and Exclusion Criteria

Randomized controlled trials (RCTs), controlled trials, intervention studies, observational studies with a control group and case-control studies that investigated the effect of combined nutritional and exercise interventions on one or more of the outcome criteria were considered eligible for inclusion. Study populations of interest were adults with CKD stages 2–5 and adults undergoing hemodialysis (HD) or peritoneal dialysis (PD) treatment. Nutritional interventions needed to include an adequate protein prescription according to dietary guidelines: dialysis patients 1.0 to 1.2 g/kg body weight and patients with CKD not on dialysis 0.55 to 0.8 g/kg body weight [27], combined with, for example, additional dietary counseling or prescribed supplementary nutrition (e.g., oral nutritional supplement (ONS), tube feeding, intradialytic parenteral nutritional (IDPN)). Exercise interventions are needed to contain exercise training, such as resistance, functional or aerobic training. Control groups receiving no intervention, nutritional intervention, exercise training, or a combined nutritional and exercise intervention of a lesser intensity were included. Studies were excluded when not based on original data (i.e., congress abstract). No restriction regarding publication year was made.

### 2.4. Outcome Measures

Outcome measures deemed relevant for this review were categorized into the following four domains [5,28]:Nutritional status;Dietary protein- and energy intake (24-h recall or dietary record),Body weight and body composition (BODPOD, dual-energy X-ray absorptiometry or bio-impedance spectroscopy),Acute phase proteins (C-reactive protein (CRP) and (pre)albumin).Muscle strength (HGS; dynamometer), knee or leg extension (one-repetition maximum (1 RM) or dynamometer) and leg press (1 RM);Physical performance (6-min walk test (6-MWT), timed up-and-go (TUG), short physical performance battery (SPPB), gait speed and sit-to-stand (STS));QoL (mental composite score (MCS) and physical composite score (PCS)).

### 2.5. Screening and Data Extraction

MdG and MD screened the yielded abstracts independently based on titles and abstracts. When articles were found to be eligible, full text papers, including the reference list, were thoroughly assessed by MdG, MD and WV. Disagreements were resolved in consultations with HK. Relevant data from the included studies were extracted using altered Cochrane data collection forms [29].

### 2.6. Risk of Bias and Quality Assessment

MdG and WV independently assessed the methodological quality of the studies included with version two of the Cochrane risk-of-bias (RoB2) tool [30].

### 2.7. Data Analysis and Statistical Methods

If three or more studies reported on the same outcome measure, the possibility of a meta-analysis was explored. Continuous data were reported as either the mean difference (MD) or standardized mean difference (SMD). The measures of effect were the differences from baseline to final follow-up. When standard deviations of these differences were missing, authors were asked to send additional data. If these data were not provided, we calculated the SD of difference with the formulas provided by Cochrane [29]. The *I*^2^ value was calculated to examine heterogeneity; if the *I*^2^ value was above 50%, a random-effects model was used instead of a fixed-effects model. Studies with multiple intervention groups, solely varying in exercise protocols, were combined in a single intervention group. If there were multiple control groups within one study, the preference was given to a control group receiving no intervention, or alternatively, a control group with either a nutritional or exercise intervention. Meta-analyses were performed in Cochrane Review Manager (RevMan) version 5.4.1 (The Cochrane Collaboration 2020) [31]. If studies had unobtainable missing data or incomparable data, or when outcomes were reported in two or fewer studies, they were qualitatively summarized.

## 3. Results

### 3.1. Study Selection

The results of the screening process are presented in a PRISMA flow diagram (Figure 1). The search strategy acquired 2682 records, and after duplicated records were removed, 1833 records remained for abstract screening. A total of 27 records were screened for the full text, of which seventeen were excluded due to not meeting the inclusion criteria (*n* = 10), study protocols (*n* = 2) or no availability of the full text (*n* = 5). The reference listing did not yield any more records. In total, nine studies (ten publications) were included [32,33,34,35,36,37,38,39,40,41].

### 3.2. Study Characteristics

#### 3.2.1. General Characteristics of Studies

The studies included data from 334 patients; details regarding study characteristics are shown in Table 1. The mean or median age ranged from 29 to 70 years. Almost all studies consisted predominantly of male participants, except for three [36,37,41]. In total, seven studies were conducted in HD or PD patients [32,33,34,36,37,38,41] and two in non-dialyzed patients with CKD [35,39,40]. The duration of the interventions ranged from three to twelve months.

#### 3.2.2. Intervention Groups

Six studies [32,34,36,37,38,41] prescribed ONS as part of the nutritional intervention, while Hristea [33] exclusively prescribed ONS in order to achieve protein and energy goals. Castaneda [39,40] aimed for a low protein diet of 0.6 g/kg, and Leehey [35] for a 200–250 caloric deficit diet.

Six studies [32,33,34,35,38,39,40] used a three times/week exercise schedule, Martin-Alemañy [36,41] either two or three times/week and Martin-Alemañy [37] two times/week. In total, five studies [32,37,38,39,40] included resistance training, two studies had aerobic training [33,34], and two studies [35,41] were a combination of both. In the Hristea [33] study, there was no mention of professional supervision during exercise, and the 40-week home training phase of Leehey [35] was not fully supervised.

#### 3.2.3. Control Groups

Differences were observed among control groups. The control group of five studies received identical nutritional interventions to the intervention group [32,33,36,37,41] and no exercise intervention. Castaneda’s [39,40] control group received low intensity stretch exercises and equal nutritional interventions to the intervention group. The control group of Jeong [34] prescribed an iso-caloric, non-protein-containing, artificially sweetened beverage. Molsted [38] prescribed the control group an identical exercise intervention to the intervention group and an isocaloric and non-protein-containing ONS.

### 3.3. Risk of Bias

The risk of bias quality assessment is shown in Figure 2. The randomization process was insufficiently described in Castaneda’s [39,40] study. Other studies scored ‘some concerns’ due to intervention providers that were not blinded [32,39,40], missing outcome data [32,35,37] and outcome assessors that were not blinded to the allocated intervention [33,34,36,37,41]. All the included RCTs [32,33,34,35,36,37,38,39,40,41] resulted in an overall assigned judgement of ‘some concerns’ due to the lack of a pre-specified analysis plan.

### 3.4. The Effect of Combined Nutritional and Exercise Interventions on Nutritional Status

#### 3.4.1. Nutritional Intake

Six studies investigated the effect of combined interventions on daily energy and protein intake [32,33,34,37,39,40], as presented in Table 2. Hristea [33] reported a mean energy increase of 11% in the intervention group (*p* = 0.03). Martin-Alemañy [37] found improved energy and protein intake within both groups (mean energy increase of 33% and protein increase of 60%, *p* not reported). Jeong [34] found an improved mean protein intake of 13% and 20% in intervention groups (*p* = 0.02) and reported no differences in energy intake.

#### 3.4.2. Body Weight and Body Composition

As presented in Table 2, five studies [32,36,37,39,40,41] included body weight, of which three [36,39,40,41] found no intergroup differences. Martin-Alemañy [37] reported slight intragroup increases in body weight (*p* < 0.05), and Dong [32] found a slight increase for the cohort as a whole (*p* = 0.02). Seven [32,33,34,35,36,37,39,40] studies investigated BMI, and the pooled analysis, including data from 229 patients (Figure 3a), showed no effect of a combined intervention (MD 0.33, 95%CI [−0.07 to 0.74], *p* = 0.11; *p* for heterogeneity < 0.01, *I*^2^ = 79%). Pooled analysis of lean body mass (LBM) (Figure 3b) of 140 patients showed no differences among the studies [32,33,34,35] (SMD −0.09, 95%CI [−0.61 to 0.42], *p* = 0.72), heterogeneity (*I*^2^ = 52%, *p* = 0.10).

#### 3.4.3. Acute Phase Proteins

Eight studies [32,33,34,35,36,37,39,40,41] reported on relevant biochemical indicators (Table 2). Regarding serum prealbumin levels, four studies [32,33,36,39] found no differences. Hristea [33] observed a slight mean increase of 3% in albumin within the intervention group *(p* = 0.03), while Martin-Alemañy [37] found a small increase in both groups (*p* not reported). The remaining four studies [32,34,39,40,41] reported no changes in albumin. Castenada [39,40] demonstrated a 25% decrease in CRP in the intervention group (*p* = 0.05), and Hristea [33] showed a 33% decrease in the intervention group (*p* not reported). However, no differences in CRP were reported in the remaining studies [32,34,35,36,41].

### 3.5. The Effect of Combined Nutritional and Exercise Interventions on Muscle Strength

Pooled analysis of 166 patients [33,34,35,38,39], presented in Figure 3c, showed no effect on knee extension (SMD 0.43, 95%CI [−0.12 to 0.97], *p* = 0.12; *p* for heterogeneity = 0.03, *I*^2^ = 63%) [33,34,35,38,39]. Regarding leg press outcomes, Castaneda [39,40] reported a larger mean increase in the intervention group of 29% than the 1% decrease in the control group (*p* = 0.001). Dong [32] reported an increased leg press of 21% in the intervention group and 11% in the control (*p* = 0.001; overall group effect). Three studies assessed HGS, of which Martin-Alemañy [37] found intragroup increases of 10% in the intervention group and 29% in the control group (*p* < 0.05). No intergroup differences were observed [36,41].

### 3.6. The Effect of Combined Nutritional and Exercise Interventions on Physical Performance

Six studies included outcomes on physical performance [33,34,35,36,38,41]. Pooled analysis of 82 patients [33,35,36], as presented in Figure 3d, showed the effect of a combined intervention on results of the 6-MWT (MD 27.2, 95%CI [7 to 47.4], *p* = 0.008), with moderate heterogeneity (*I*^2^ = 45%, *p* = 0.16). Four studies included TUG [34,35,36,41]. A meta-analysis of 129 patients, as presented in Figure 3e, showed no pooled effect (MD 30, 95%CI [−0.73 to −0.13, *p* = 0.17), with low heterogeneity (*I*^2^ = 7%, *p* = 0.34). Other studies assessed physical performance with the shuttle walk test (SWT) [34], gait speed [34,41], STS [34,36,41], and chair stand test [38]. No effect of combined intervention on these outcomes was reported.

### 3.7. The Effect of Combined Nutritional and Exercise Interventions on QoL

Four studies [33,34,35,38] included QoL measured with the short form-36 (SF-36), which was subdivided into MCS and PCS. A meta-analysis of 136 patients showed no differences between intervention or control regarding the MCS (Figure 3f) (MD 6.24, 95%CI [−2.64 to 15.11], *p* = 0.17; *p* for heterogeneity < 0.00001, *I*^2^ = 94%) and PCS (Figure 3g) (MD 3.39, 95%CI [−0.92 to 7.69], *p* = 0.12; *p* for heterogeneity < 0.00001, *I*^2^ = 91%). Martin-Alemañy [36,37,41] measured QoL with the KDQOL-SF and reported no intergroup differences.

## 4. Discussion

In this systematic review and meta-analysis, we analyzed data from nine studies that implemented combined nutritional and exercise interventions in the care of patients with CKD. Our findings indicate that combined interventions were effective in improving the 6-MWT. No significant effects were found on other subdomains of nutritional status, muscle strength, other physical performance tests or QoL.

### 4.1. Limitations

Several factors may have influenced the results, as there were concerns about the methodological quality of the studies included. These concerns mainly revolved around uncertainties regarding missing outcome data, randomization and blinding of participants and personnel. While six studies reported on sample size calculations [32,33,34,35,36,41], with the aim of achieving 80% to 90% power to detect differences, only four studies [32,34,36,41] included the required number of patients at baseline. Notably, high dropout rates were reported in these studies, as Dong [32] reported a 32% dropout rate, Jeong [34] 41% and 24% for Martin-Alemañy [36]. Two studies did not reach their initial inclusion target [33,35] or did not report on sample size calculations [37,38,39,40]. Some studies [33,35,38] inadequately described the dropouts and adherence to the intervention or control regimen. High dropout rates can limit the generalizability of the findings, as CKD patients who volunteer for exercise interventions may be biased towards the healthier subjects. Considering that all included studies were underpowered to detect significant intergroup differences, it is not possible to draw robust conclusions about the effects of combined nutritional and exercise interventions in patients with CKD.

The studies included in this review were extremely heterogeneous in terms of interventions and study outcomes. The nutritional interventions varied in dosage, duration, frequency of administration and timing. Similarly, substantial differences were observed in exercise protocols, including variations between resistance and aerobic exercises, as well as differences in duration and intensity. In order to evaluate the influence of different protein recommendations between dialysis and non-dialyzed patients, two post hoc analyses were performed, excluding non-dialyzed patients. The results pertaining to BMI (MD 0.11, *p* = 0.53) and knee extension (MD 0.28, *p* = 0.25) demonstrated similar results. Additionally, another aspect that might influence the results of nutritional and exercise interventions is the prevalence of diabetes among the CKD participants. Of the nine studies included, two studies consisted of 40% to 45% of diabetes patients and one study contained all diabetes patients. The other remaining studies included either none or a maximum of 20% of diabetes patients in their study group. Since there was a small number of diabetes patients in all studies, no additional analyses were performed.

Also, the control groups displayed a wide range of characteristics. Only one study [34] included a ‘usual care’ group without a nutritional and exercise intervention. Some control groups showed increases in energy and protein intake, which could have influenced the results as well. Three studies [32,33,37] showed an increased energy and protein intake in the control groups, and one study [39,40] showed an increase in the control group’s energy intake only. Additionally, the results of two [35,37] studies were of limited value, as they solely relied on intragroup statistical comparisons. Therefore, identifying the most effective intervention proved to be challenging.

### 4.2. Effect of Combined Nutritional and Exercise Interventions on Lean Body Mass and Muscle Strength

The pooled analysis did not demonstrate any impact on LBM and muscle strength, which could be attributed to the overstimulation of catabolic pathways in patients with CKD. These pathways are associated with complications such as insulin resistance and the accumulation of metabolic waste products, potentially hindering the interventions’ effectiveness [42]. It is plausible that specific protein goals and exercise type, intensity or duration are required to counteract this catabolic state. The prescribed interventions may not have adequately addressed these requirements. The amount of protein prescribed varied considerably among studies, including low protein diets [39,40], no specific protein recommendations [35] and offering ONS ranging from 9 g/day up to 38 g/day [32,34,41].

Unfortunately, among the four studies [32,33,34,35] included in the pooled analysis on LBM, only one study [32] focused exclusively on resistance exercise. Resistance training is well known for its ability to promote muscle growth and strength [43], unlike aerobic exercise, which primarily impacts cardiorespiratory fitness [44]. In terms of strength, the meta-analysis, including five studies, demonstrated that the combined intervention had no effect on knee extension. Among these five studies [33,35,38,39,40], only three studies [35,38,39,40] incorporated resistance exercise as part of the intervention, whereas the other two studies offered solely aerobic exercise [33,34].

### 4.3. Effect of Combined Nutritional and Exercise Interventions on Physical Performance

Six studies investigated physical performance using a range of tests [33,34,35,36,38,41]. The meta-analysis, including data from three [33,35,36] of these studies, found a favorable effect of the combined intervention on the 6-MWT, while no effect was found on the TUG test. This contrast could be due to the fact that the 6-MWT is a functional test that measures endurance at a submaximal level, whereas the TUG test primarily focuses on lower extremity strength [45]. It is plausible that the offered interventions had a more immediate impact on endurance rather than strength.

### 4.4. Effect of Combined Nutritional and Exercise Interventions on QoL

It is recognized that nutritional status and physical performance are important factors contributing to health-related QoL [46,47,48]. Our meta-analysis, including four studies [33,34,35,38], showed no pooled effect on MCS and PCS. The lack of improvement in QoL scores may be due to the variability in outcomes among studies, with only low to moderate improvements in some components of nutritional status, muscle strength and physical performance. Additionally, patients with CKD have a high disease burden, and the effect of combined nutritional and exercise interventions may not be sufficient to address all health concerns.

## 5. Conclusions

In conclusion, the limited statistical power and large heterogeneity of the included studies raised concerns about the reliability and generalizability of the findings. The current knowledge gaps highlight the absence of sufficient evidence upon which to base recommendations. Therefore, future research should prioritize conducting high quality and homogenous trials. These trials should focus on developing personalized interventions that can enhance long-term adherence, ultimately aiming to create sustainable changes in patients’ daily lifestyles. Additionally, we suggest carefully aligned measurement methods according to the intervention given. By doing so, future studies can contribute to a more comprehensive understanding of the impact of combined interventions in improving PEW/malnutrition-related outcomes for patients with CKD.

## Figures and Tables

**Figure 1 nutrients-16-00406-f001:**
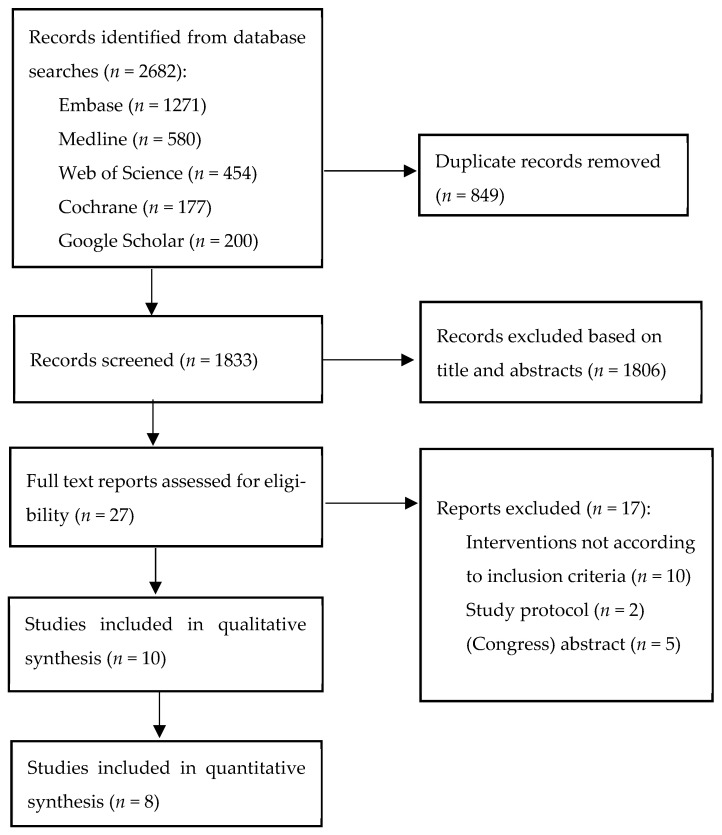
Flow diagram of selection of studies on combined exercise and nutritional intervention in chronic kidney disease patients.

**Figure 2 nutrients-16-00406-f002:**
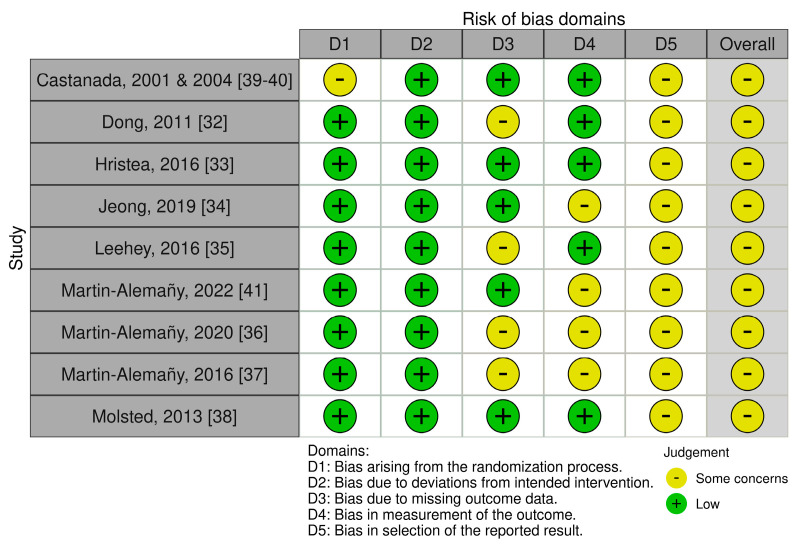
Risk of bias assessment of the included randomized controlled trials [32,33,34,35,36,37,38,39,40,41].

**Figure 3 nutrients-16-00406-f003:**
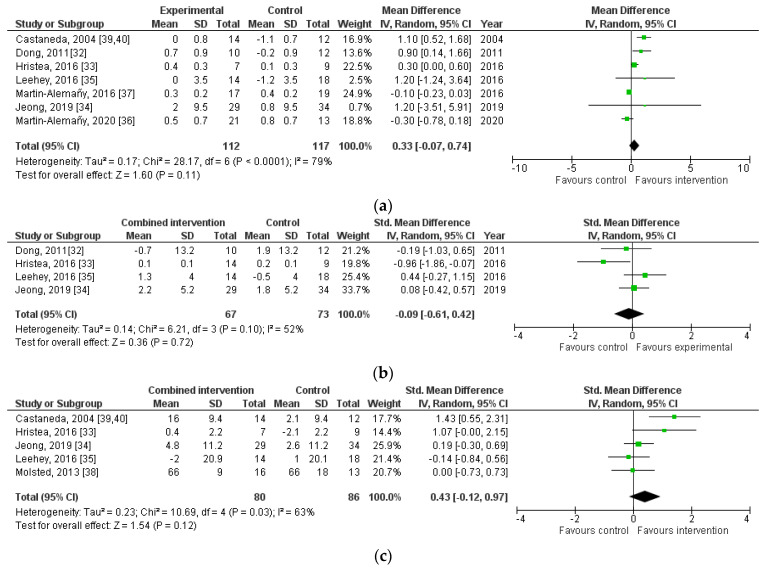
Forest plots comparing the effects of combined nutritional and exercise interventions on nutritional status, muscle strength, physical performance, and quality of life, assessed with (**a**) Body mass index; (**b**) Lean body mass; (**c**) Knee extension; (**d**) Six-minute walk test; (**e**) Timed up-and-go test, (**f**) Mental component score; (**g**) Physical component score [32,33,34,35,36,37,38,39,40].

**Table 1 nutrients-16-00406-t001:** Study characteristics of the included studies on combined nutritional and exercise interventions in patients with CKD.

First Author, Year	Study Design	Trial Duration (mo), Sample Size	Study Participants, Mean Age	Methods of Intervention			Relevant Outcomes Measures			
				Nutrition	Exercise		Nutritional Status	Muscle Strength	Physical Performance	QoL
					Type	Frequency				
Castaneda, 2001 & 2004 [39,40]	RCT	3 I: 14 C: 12	CKD, non-dialysis 65 ± 10 y	I: Supervised LPD 0.6 g/kg C: Same as intervention	I: Supervised resistance training C: Supervised low-intensity exercises	I: 3 times/wk C: 3 times/wk	Protein intake Energy intake Body weight BMI CRP Albumin Pre albumin	Leg press Knee extension	Not reported	Not reported
Dong, 2011 [32]	RCT	6 I: 15 C: 17	HD (3 times/wk) 43 ± 13 y	I: ONS (2 × 480 kcal, 17 g protein) taken prior to, during or after HD C: Same as intervention	I: Supervised resistance training C: No exercise	I: 3 times/wk	Protein intake Energy intake BMI LBM CRP Albumin Pre albumin	Leg press	Not reported	Not reported
Hristea, 2016 [33]	RCT	6 I: 7 C: 9	HD (3 times/wk) diagnosed with PEW 70 ± 15 y	I: Dietary counseling by a dietitian (aiming 30–40 kcal/kg and >1.1 g/kg ideal weight/day) C: Same as intervention	I: Supervised aerobic training C: No exercise	I: 3 times/wk	Protein intake Energy intake BMI LTI CRP Albumin Pre albumin	Knee extension	6-MWT	SF-36 PCS SF-36 MCS
Jeong, 2019 [34]	RCT	12, I1: 38 I2: 29 C: 34	HD (3 times/wk) 55 ± 12 y	I1: 30 g whey protein supplement during HD I2: Same as intervention 1 C: ±150 g of non-caloric, non-protein containing beverage during HD	I1: No exercise I2: Supervised aerobic training C: No exercise	I2: 3 times/wk	Protein intake Energy intake BMI Lean mass Albumin CRP	Knee extension	SWT TUG Normal gait speed STS	SF-36 PCS SF-36 MCS
Leehey, 2016 [35]	RCT	12 I: 18 C: 18	CKD (stage 2–4) with type 2 diabetes and BMI > 30 m^2^/kg 66 ± 8 y	I: Dietary counseling at baseline aiming for a 200–250 calorie deficit with 9 follow-up calls C: Same as intervention	I: Resistance and aerobic training C: No exercise	I: 3 times/wk	Lean body weight BMI CRP	Knee extension	6-MWT TUG	SF-36 PCS SF-36 MCS
Martin-Alemañy, 2022 [41]	RCT	24 I: 10 C: 14	HD (2–3 times/wk) 34 ± 11	I: ONS (2 × 434 kcal, 19 g protein) C: ONS (2 × 434 kcal, 19 g protein)	I: Supervised resistance and aerobic training C: No exercise	I: 2–3 times/wk	Body weight CRP Albumin	HGS	6-MWT TUG STS Gait speed	KDQOL-SF
Martin-Alemañy, 2020 [36]	RCT	3 I1: 9 I2: 12 C: 13	HD (2–3 times/wk) 29 ± 9	I1: ONS (1 × 480 kcal, 20 g protein) taken during HD I2: Same as intervention C: Same as intervention	I1: Supervised resistance training I2: Supervised aerobic training C: No exercise	I1: 2–3 times/wk I2: 2–3 times/wk	Body weight BMI CRP Albumin	HGS	6-MWT TUG STS	KDQOL-SF
Martin-Alemañy, 2016 [37]	RCT	3 I: 17 C: 19	HD (2 times/wk) 34 (25–43)	I: ONS (1 × 430 kcal, 19 g protein) before and during HD C: Same as intervention	I: Supervised resistance training C: No exercise	I: 3 times/wk	Protein intake Energy intake Body weight BMI Albumin	HGS	Not reported	KDQOL-SF
Molsted, 2013 [38]	RCT	4 I: 16 C: 13	HD (24), PD (5) 55 ± 14	I: ONS (1 × 250 kcal, 9 g protein, 25 g carbohydrates) C: ONS (1 × 250 kcal, 0 g protein, 2 g carbohydrates)	I: Supervised resistance training C: Same as intervention	I: 3 times/wk C: 3 times/wk	Not reported	Knee extension right	CTS	SF-36 PCS SF-36 MCS

Abbreviations: 6-MWT: six-minute walk test, BIA: bio-impedance analysis, BMI: body mass index, C: control group, CKD: chronic kidney disease, CRP: C-reactive protein, CST: chair stand test, HD: hemodialysis, HGS: handgrip strength, I: intervention group, LBM: lean body mass, LTI: lean tissue index, LPD: low protein diet, MCS: mental component scale, ONS: oral nutritional supplements, PCS: physical component scale, PEW: protein-energy wasting, STS: sit to stand test, SWT: shuttle walk test, TUG: timed up-and-go test, QoL; quality of life. Data is presented as mean ± SD or SED or median.

**Table 2 nutrients-16-00406-t002:** Summary of nutritional status outcomes reported in the included studies.

Author	Outcomes			Results *		
		Intervention		Control		*p*-Value
		Baseline	Last Follow-Up	Baseline	Last Follow-Up	
Castenada [39,40]	Protein intake (g/kg)	0.6 ± 0.1	0.6 ± 0.1	0.7 ± 0.1	0.6 ± 0.1	*p* > 0.2
	Energy intake (J/kg)	68 ± 27	76 ± 32	87 ± 28	98 ± 25	*p* > 0.2
	Body weight (kg)	85 ± 16	85 ± 16	76 ± 14	73 ± 9	*p* = 0.05
	CRP (mg/L)	8 ± 6	6 ± 6	6 ± 6	8 ± 6	*p* = 0.05 (group effect)
	Albumin (g/dL)	4 ± 0.3	4 ± 0.2	4 ± 0.4	4 ± 0.4	*p* = 0.09
	Prealbumin (mg/dL)	253 ± 46	276 ± 42	232 ± 60	234 ± 50	*p* = 0.05
Dong [32]	Protein intake (g/kg)	0.8 ± 0.2	1.0 ± 0.3	0.8 ± 0.3	1.1 ± 0.4	*p* > 0.05
	Energy intake (kcal/kg)	24 ± 7	27 ± 7	22 ± 9	28 ± 12	*p* > 0.05
	Body weight (kg)	76 ± 15	75 ± 13	84 ± 17	86 ± 21	*p* = 0.02 (↑ overall time effect group)
	CRP (mg/L)	4 (2–13)	3 (1.3–8.3)	4 (1–12)	7 (6–12)	*p* > 0.05
	Albumin (mg/L)	41 ± 3	42 ± 4	42 ± 3	42 ± 2	*p* > 0.05
	Prealbumin (mg/dL)	40 ± 11	42 ± 12	38 ± 10	42 ± 7	*p* > 0.05
Hristea [33]	Protein intake (g/kg)	1.1 ± 0.2	1.2 ± 03	0.9 ± 0.2	1.2 ± 0.4	*p* = 0.01 (overall time effect group)
	Energy intake (kcal/kg)	27 ± 4	30 ± 7	21 ± 4	28 ± 8	*p* = 0.03 (main group effect, I ↑)
	CRP (mg/L)	6 ± 8	2 ± 2	6 ± 6	5 ± 6	Not reported
	Albumin (mg/L)	38 ± 3	39 ± 3	40 ± 4	39 ± 4	*p* = 0.03 (time*group, I ↑, C ↓)
	Prealbumin (g/L)	226 ± 45	232 ± 27	251 ± 69	227 ± 56	Not reported
Jeong [34]	Protein intake (g/kg)	0.8 ± 0.5 0.8 ± 0.4	0.9 ± 0.3 (I1) 1.0 ± 0.5 (I2)	0.8 ± 0.4	0.7 ± 0.4	*p* = 0.02 (time*group)
	Energy intake (kcal/kg)	18 ± 8 17 ± 7	19 ± 9 (I1) 20 ± 12 (I2)	19 ± 11	17 ± 8	*p* = 0.16 (time*group)
	CRP (mg/L)	18 ± 21 15 ± 14	11 ± 8 (I1) 13 ± 12 (I2)	7 ± 6	11 ± 11	*p* = 0.40 (time*group)
	Albumin (g/L)	4 ± 0.4 4 ± 0.4	4 ± 0.3 (I1) 4 ± 0.5 (I2)	4 ± 0.3	4 ± 0.3	*p* = 0.71 (time*group)
Leehey [35]	CRP (mg/L)	6 ± 8	8 ± 14	9 ± 11	7 ± 8	*p* = 0.23
Martin-Alemañy [41]	Body weight (kg)	56 ± 9	58 ± 9	55 ± 7	56 ± 7	*p* = 0.46 (time*group)
	CRP (mg/L)	5 (1–13)	3 (3–9)	6 (3–9)	4 (2–7)	*p* = 0.78 (time*group)
	Albumin (g/dL)	4 ± 0.4	4 ± 0.5	4 ± 0.5	4 ± 0.3	*p* = 0.4 (time*group)
Martin-Alemañy [36]	Body weight (kg)	53 ± 6 52 ± 9	55 ± 5 (I1) 53 ± 8 (I2)	52 ± 10	53 ± 9	*p* = 0.22 (time*group)
	CRP (mg/L)	3 (3–9) 7 (3–13)	6 (4–9) (I1) 5 (3–17) (I2)	4 (2–4)	3 (2–6)	*p* = 0.44 (time*group)
	Albumin (g/L)	4 ± 0.5 4 ± 0.3	4 ± 0.4 (I1) 3 ± 0.5 (I2)	4 ± 0.5	4 ± 0.4	*p* = 0.42 (time*group)
Martin-Alemañy [37]	Protein intake (g/kg)	1.0 ± 0.4	1.6 ± 0.5	1.0 ± 0.6	1.6 ± 0.5	↑ (no *p* reported)
	Energy intake (kcal/kg)	27 ± 11	36 ± 15	27 ± 11	35 ± 16	↑ (no *p* reported)
	Body weight (kg)	51 (46–57)	52 (47–58)	47 (43–52)	49 (45–54)	↑ (no *p* reported)
	Albumin (g/dL)	3 ± 0.3	4 ± 0.3	4 ± 0.3	4 ± 0.4	↑ (no *p* reported)

Abbreviations: Alb: albumin, BW: body weight, CRP: C-reactive protein, I1: intervention group 1, I2: intervention group 2. * Results are presented with baseline values and values of the last follow-up measurement of a study, with mean ± SD or SED or median (IQR).

## Data Availability

Not applicable.

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
