# Peer review of "The Impact of Combined Nutrition and Exercise Interventions in Patients with Chronic Kidney Disease"

_nutrients, 2024, doi:10.3390/nu16030406_

Round 1

Reviewer 1 Report

Comments and Suggestions for Authors

Major comment 

Although the authors selected 9 studies for the systemic review and meta-analysis based on clinical design, several studies showed apparent changes in nutritional markers during study periods, It seems to me that the conclusion drawn from the absence of significant inter-group differences did not support the absent effect of nutrition and exercise intervention in CKD patients.  Furthermore, since the prevalence of diabetes might affect those effect in CKD patients, it would be better to restrict these analyses to those with or without diabetes.

Minor comments

Line 82. Why authors included CKD stage 2 patients, because malnutrition/PEW and sarcopenia is assumed not overt.

Diabetes is known as a major cause for malnutrition/PEW and sarcopenia.  It would be desired to know the effect of diabetes complication on CKD induced malnutrition/PEW and sarcopenia.

Body weight and lean body mass did not differ significantly between control and intervention group.  Since intervention is known to affect muscle strength and muscle quality more sensitively than muscle mass and lean body mass, it would be interesting to analyze the effect of intervention on muscle strength and muscle quality rather than muscle mass and lean body mass.

Since several control groups showed apparent increase of energy intake (Castenada, Dong, Hristea, Martin-Alemany), protein intake(Dong, Hristea, , Martin-Alemany), it would be difficult to draw the conclusion indicating the absence of the positive effect of combined nutrition and exercise intervention in CKD patients. It is reasonable that control patients should maintain the same body weight, and protein/calorie intake during the study periods.

In line 225, the authors mentioned significant increase of HGS by 29% in the control group, in whom apparent increases of protein intake, calorie intake, and BW were found (Table 2).  Was their HGS increase responsible for those increases in nutritional markers?  So, if nutritional improvement in them might contribute to the improvement of HGS, it seems to me that the nutritional interventions might contribute to HGS increases.

Reviewer 2 Report

Comments and Suggestions for Authors

In this study, the authors reviewed the effect of combined exercise and nutrition on clinical outcomes using the 9 published RCT studies. They showed that combined nutrition and exercise therapy improved 6-MWT, while did not nutritional status, muscle power or QOL scores. This is an interesting analysis, while there are some concerns.

1.     In this study, the authors included 2 RCTs which participants were pre-dialysis CKD. However, the recommended dietary protein intake was quite different between pre-dialysis and dialysis patients. So, the authors should re-analyze using 7 RCTs with those on maintenance dialysis.  

2.     Trial duration was ranged from 3 to 24 months. Did you check the adherence to study protocol in each trial?

3.     The authors compared the effects of combined nutrition exercise intervention on anthropometric nutritional parameters such as BMI and lean body mass. Did you examine the impact of combined therapy on biochemical nutritional markers such as transthyretin and albumin?

Round 2

Reviewer 1 Report

Comments and Suggestions for Authors

The revised version has now been improved enough for publication in NUTRIENTS.